# Octoploids Show Enhanced Salt Tolerance through Chromosome Doubling in Switchgrass (*Panicum virgatum* L.)

**DOI:** 10.3390/plants13101383

**Published:** 2024-05-16

**Authors:** Jiali Ye, Yupu Fan, Hui Zhang, Wenjun Teng, Ke Teng, Juying Wu, Xifeng Fan, Shiwen Wang, Yuesen Yue

**Affiliations:** Institute of Grassland, Flowers and Ecology, Beijing Academy of Agriculture and Forestry Sciences, Beijing 100097, China; 1 jialiye@nwafu.edu.cn (J.Y.); fanyupu2024@163.com (Y.F.); peakzeal@163.com (H.Z.); tengwenjun@126.com (W.T.); tengke.123@163.com (K.T.); wujuying@grass-env.com (J.W.); fanxifengcau@163.com (X.F.); 2State Key Laboratory of Soil Erosion and Dryland Farming on the Loess Plateau, Institute of Soil and Water Conservation, Northwest A&F University, Yangling, Xianyang 712100, China; 3College of Natural Resources and Environment, Northwest A&F University, Yangling, Xianyang 712100, China

**Keywords:** *Panicum virgatum* L., octoploid, salt stress, transcriptome, WGCNA

## Abstract

Polyploid plants often exhibit enhanced stress tolerance. Switchgrass is a perennial rhizomatous bunchgrass that is considered ideal for cultivation in marginal lands, including sites with saline soil. In this study, we investigated the physiological responses and transcriptome changes in the octoploid and tetraploid of switchgrass (*Panicum virgatum* L. ‘Alamo’) under salt stress. We found that autoploid 8× switchgrass had enhanced salt tolerance compared with the amphidiploid 4× precursor, as indicated by physiological and phenotypic traits. Octoploids had increased salt tolerance by significant changes to the osmoregulatory and antioxidant systems. The salt-treated 8× Alamo plants showed greater potassium (K^+^) accumulation and an increase in the K^+^/Na^+^ ratio. Root transcriptome analysis for octoploid and tetraploid plants with or without salt stress revealed that 302 upregulated and 546 downregulated differentially expressed genes were enriched in genes involved in plant hormone signal transduction pathways and were specifically associated with the auxin, cytokinin, abscisic acid, and ethylene pathways. Weighted gene co-expression network analysis (WGCNA) detected four significant salt stress-related modules. This study explored the changes in the osmoregulatory system, inorganic ions, antioxidant enzyme system, and the root transcriptome in response to salt stress in 8× and 4× Alamo switchgrass. The results enhance knowledge of the salt tolerance of artificially induced homologous polyploid plants and provide experimental and sequencing data to aid research on the short-term adaptability and breeding of salt-tolerant biofuel plants.

## 1. Introduction

Switchgrass is a C_4_ warm-season perennial species native throughout North America that exhibits strong environmental adaptability [1,2]. The US Department of Energy has selected switchgrass as a preferred feedstock for the production of cellulosic biofuels [3,4]. Its broad adaptability and rapid growth rate provide a stable and ample supply of biomass for biofuel production [5]. As an important raw material for ethanol bioenergy, switchgrass is considered an ideal plant to alleviate soil salinization and grows well in alkaline saline and marginal soil [6].

Soil salinization severely restricts global agricultural production and ecological restoration [7]. Salt stress can be categorized into two stages: the osmotic stress stage in which plant growth decreases rapidly because of the high external osmotic pressure in the early phase, and the ion toxicity stage in which plant growth is slowly inhibited owing to the abundance of sodium ions (Na^+^) that accumulate in the leaves in the advanced phase [8]. To combat the harmful effects of salt stress, plants have evolved a number of regulatory strategies, including physiological, biochemical, and morphological responses. For example, plants balance the osmotic pressure and maintain the intracellular ion balance through the osmotic regulation of inorganic ions, such as Na^+^, K^+^, and chlorine (Cl^−^), and/or organic molecules, such as proline, soluble sugar, and protein [9]. Furthermore, the antioxidant enzyme system of plants plays a role in maintaining the physiological functions of cells by increasing the expression of relevant genes or the activity of antioxidant enzymes to detoxify reactive oxygen species (ROS) in cells [10]. In addition, the alkaline salt tolerance of switchgrass may be achieved by the regulation of ion homeostasis, transport proteins, detoxification, heat shock proteins, dehydration, and sugar metabolism [7,11].

Polyploidy or whole-genome duplication pertains to an organism or cell that contains two or more sets of chromosomes. Polyploidization arises from either genome duplication or the combination of different genomes of related species [12]. Chromosome doubling in artificially induced autopolyploids is mainly achieved by the inhibition of the formation of spindle filaments by treatment with mutagens, such as colchicine [13]. Accumulating evidence suggests that polyploidization has been a common phenomenon during the evolution of angiosperms and confers potential advantages in fitness under unfavorable environments [14,15,16]. Previous studies have demonstrated that polyploids are generally more tolerant of abiotic stresses, including drought [17], chilling [18], and salt [19].

Research on the salt tolerance of autopolyploid plants has mainly been focused on the physiological level [20,21,22,23]. Plants can better maintain homeostasis under stress by regulating the cell size, structure, and membrane system [24]. On the one hand, polyploid plants have improved adaptability through enhanced accumulation of certain beneficial substances, such as soluble sugar, proline, and antioxidant enzymes, in response to salt stress [25,26]. In addition, polyploidy may reduce the content of harmful substances, such as malondialdehyde (MDA) and Na^+^, maintain the dynamic balance of ROS, reduce oxidative damage to cells, and thereby enhance salt tolerance [27,28]. The gene regulatory networks of polyploid plants responding to salt stress are diverse, including the biosynthesis of osmotic regulators and stress tolerance proteins, ion transport, ion balance genes, and the expression of stress-related transcription factors [29,30].

Previous studies have achieved substantial progress in research on the mechanism of salt tolerance of switchgrass [31]. It has been found that switchgrass is most sensitive to salt at the seedling stage [3], and the salt tolerance of switchgrass varies from variety to cultivar [32,33]. Under high salinity, switchgrass can quickly regulate stomatal conductance, osmotic regulation, and compartmentalization of Na^+^ [34]. Salt stress affects lipid peroxidation and antioxidant defense of switchgrass leaves and roots [35]. In a previous study, we treated tetraploid switchgrass callus with colchicine, then used flow cytometry to screen the ploidy of regenerated plants and ultimately obtained octoploid switchgrass [36]. The present study focused on the differences in physiological and biochemical characteristics in leaves of tetraploid and octoploid switchgrass under salt stress. In addition, an RNA-sequencing (RNA-seq) of the root transcriptome was performed to analyze the mechanisms of octoploid switchgrass responses to salt stress. The results provide a theoretical foundation for the popularization and application of octoploid switchgrass.

## 2. Results

### 2.1. Phenotypic and Physiological Changes in Tetraploid and Octoploid Switchgrass in Response to Salt Stress

The phenotypes of 4× and 8× individuals differed notably before and after salt-stress treatment, with the main stress symptoms of leaf wilting and necrosis observed (Figure 1). Octoploid plants were shorter than tetraploids before treatment but were taller than tetraploids following salt treatment. The leaves of 4× plants became yellowed and wilted, and plant growth was severely inhibited. The plant height, stem diameter, leaf length, leaf width, and biomass of 4× plants under salt stress were significantly decreased by 37.10%, 19.04%, 25.00%, 16.13%, and 54.16%, respectively, compared with those of the control (Table 1). Similarly, in 8× plants, these growth indicators were significantly decreased by 24.73%, 20.33%, 23.47%, 20.00%, and 43.30%, respectively, compared with those of the control. Under salt stress, the plant height, stem diameter, leaf length, leaf width, and biomass of 8× plants were significantly higher than those of 4× individuals by 9.50%, 16.26%, 28.48%, 23.08%, and 16.18%, respectively. Thus, 4× switchgrass was more severely affected by salt stress than were 8× plants.

Under the control condition, the total chlorophyll content in the leaves of 8× individuals was significantly lower than that in the leaves of 4× plants. Under salt stress, the chlorophyll content of 4× leaves decreased by 11.27% compared with the control, whereas that of 8× leaves was increased by 29.39% (Figure 2A).

Soluble sugars and proline are osmoregulatory substances. During salt stress, soluble sugars reduce the cell osmotic potential to maintain cell turgor and protect against the dehydration of the cells. The soluble sugar content of 8× leaves was significantly lower than that of 4× leaves (*p* < 0.01) in the control. However, the soluble sugar content of 4× leaves decreased under salt stress, whereas that of 8× leaves increased by 27.52%, compared with that of the control (Figure 2B). Generally, under adverse environments, proline gradually accumulates in plants, which plays a positive role in regulating the plant response to salt stress, thereby improving the stress tolerance of the plant. The proline content of control 8× leaves was lower than that of 4× leaves, but that of 8× plants under salt stress was 26.97% higher than that of 4× plants (*p* < 0.05) (Figure 2C).

Changes in ROS concentrations are often associated with plant response to abiotic stresses. To determine whether ploidy affected changes in the antioxidant capacity associated with exposure to salt stress, we measured stress-related physiological indicators of 4× and 8× plants in the control and salt-stress treatments for 7 days. The activities of the ROS-scavenging enzymes SOD and CAT were comparable in the 4× and 8× plants in the absence of salt stress. In response to salt stress, the activities of SOD and CAT increased in 8× plants by 7.90% and 8.60% to the control, respectively. Under salt stress, the accumulation of ROS in plants causes the formation of MDA, and the functioning of cell membranes is disrupted. The MDA content of 4× plants under salt stress increased by 18.33%, whereas that of 8× plants decreased by 14.27%, indicating that the degree of membrane lipid peroxidation and the severity of membrane damage were greater in tetraploids (Figure 2D–F).

Under salt stress, the increase in Na^+^ content affects the absorption of essential elements by plants and causes physiological disorders as a result. As an important inorganic solute, K^+^ is essential for the reduction of the plant cell osmotic potential and maintaining water balance [37]. Iron homeostasis and water uptake are maintained by low Na^+^ uptake and a high K^+^ content under saline conditions [38]. Under salt stress, the increase in Na^+^ affects the absorption of essential elements in plants and causes physiological disorders in plants. As an important inorganic solute, K^+^ is essential for reducing plant cell osmotic potential and maintaining water balance [37]. Physiologically, iron homeostasis and water uptake are maintained by low Na^+^ uptake and a high K^+^ under saline conditions [38]. The K^+^ content of 8× was significantly increased by 15.00% more than that of 4× under salt stress. The Na^+^ content of 8× and 4× showed no significant difference under salt stress (Figure 2G–I).

These results indicated that 8× switchgrass coped better with oxidative stress damage by reducing the MDA content and increasing the contents of chlorophyll and osmotic adjustment substances (soluble sugars and proline) and antioxidant enzyme activity (SOD and CAT). In addition, compared with 4× plants, 8× switchgrass showed superior salt tolerance by accumulating greater amounts of K^+^ and increasing the K^+^/Na^+^ ratio.

### 2.2. Transcriptome Sequencing

We filtered the raw RNA-seq data and obtained 79.29 Gb clean reads. Overall, the GC content ranged between 51.71% and 52.79%, with >92.69% bases having a quality score of Q30 or higher (Appendix A), indicating that the sequencing data were of high quality and sufficiently reliable for further analysis. The clean reads for each sample were aligned with the switchgrass reference genome. The final comparison efficiency was 91.09%, and the single matching rate was 87.23%. The matching rate was 3.64% (Appendix A), indicating that the output data were adequate for further analysis and ensured the reliability of subsequent data analysis.

### 2.3. Identification of Differentially Expressed Genes

To identify salt-responsive genes that were differentially expressed in the roots of 4× and 8× plants, four comparisons (4S vs. 4C, 8S vs. 8C, 8C vs. 4C, and 8S vs. 4S) were performed for the analysis of differentially expressed transcripts (DETs). A total of 1820 DEGs showed a differential expression pattern in roots of salt-treated 4× plants compared with control plants (4S vs. 4C), of which 933 genes were upregulated and 877 were downregulated. In contrast, 1354 DEGs were detected in roots of 8× plants (8S vs. 8C), of which 443 genes were upregulated and 911 were downregulated. In the comparison between salt-treated 4× and 8× plants (8S vs. 4S), 50 DEGs were identified in 8× roots, of which 31 were upregulated and 19 were downregulated (Figure 3A,B). Among these genes, 23 DEGs were expressed uniquely in 8× plants under salt stress (Figure 3B,C).

### 2.4. Go and KEGG Pathway Analysis

To gain insight into the potential mechanisms that distinguish the responses of 8× and 4× plants to salt stress, we conducted a GO enrichment analysis of the DEGs detected in 8× plants relative to those of 4× plants under salt stress by comparing the transcriptomes of 4S vs. 4C and 8S vs. 8C. The most enriched GO terms for 971 DETs specific to 4× plants comprised oxidoreductase activity (GO: 0016491, 121 genes), oxidation-reduction process (GO: 0055114, 115 genes), heme binding (GO: 0020037, 59 genes), and tetrapyrrole binding (GO: 0046906, 59 genes). In 8× plants, the most enriched GO functions were heme binding (GO: 0020037, 29 genes), tetrapyrrole binding (GO: 0046906, 29 genes), oxidoreductase activity (GO: 0016491, 55 genes), and peroxidase activity (GO: 0004601, 14 genes). The DEGs of 4× and 8× plants under salt stress were specifically enriched in oxidoreductase activity. The response of 8× plants to salt stress was also associated with peroxidase activity. Therefore, notable differences were detected in the responses of 8× and 4× plants to salt stress. Salt stress also promoted the differential expression of genes involved in the oxidation-reduction process (Figure 4A–D).

To identify metabolic pathways potentially associated with the enhanced salt tolerance of 8× plants, we performed a standard KEGG pathway enrichment analysis. We analyzed the genes that were upregulated and downregulated in the octoploids relative to those in the tetraploids under salt stress based on a pairwise comparison (4S vs. 4C and 8S vs. 8C) and evaluated the 10 most significantly enriched pathways (Figure 4E,F). The upregulated and downregulated genes under salt stress were enriched in five metabolic pathways: biosynthesis of secondary metabolites, metabolic pathways, MAPK signaling pathway-plant, phenylpropanoid biosynthesis, and plant hormone signal transduction. Thus, these metabolic pathways were indicated to play an important role in improving the salt tolerance of 8× plants.

### 2.5. Hormonal Signaling Is Altered in Octoploids under Salt Stress

Considering that plant hormones play vital roles in abiotic stress responses [28], we focused on the plant hormone signal transduction pathway and compared the expression of the relevant genes in the octoploids with that in the tetraploids under the control and salt-stress treatment. To investigate the plant endogenous hormone signal transduction in 8× switchgrass in response to salt stress, we analyzed the hormone signal transduction pathway (Figure 5). Under salt stress, a total of 31 DEGs were enriched in the plant signal transduction pathways. The enriched DEGs were associated with auxin, abscisic acid (ABA), cytokinin, and ethylene signaling. Among these phytohormones, auxin exerts its regulatory role, at least to a certain extent, by controlling the fundamental processes of cell division, expansion, and differentiation. The auxin pathway includes the AUX1/IAA, SAUR, and GH3 gene families. SAUR auxin receptor-related genes were upregulated in each treatment, whereas the auxin receptor protein-related genes AUX1/IAA and GH3 were downregulated (Figure 5A). In the ABA pathway, the receptor protein PYR/PYL-related genes were upregulated, whereas PP2C-, SnRK2-, and ABF-related genes were downregulated (Figure 5B). The expression of A-ARR, which is associated with cytokinin signaling, was downregulated under salt stress (Figure 5C). In the ethylene pathway, EIN3 receptor protein-related genes were downregulated (Figure 5D).

### 2.6. WGCNA Module Generation and Functional Enrichment Analysis

To construct a specific phenotype-related gene regulatory network, the DEGs were selected for WGCNA after filtering. To ensure that the gene distribution conformed to the scale-free network, we determined that the power was 18 (Appendix A). In addition, the correlation between the expression of each gene and the eigenvalues of the module was analyzed. Correlation analysis among modules shows that the modules with very high correlation are within the same module, which proves that the gene expression patterns among modules are similar, and there are significant differences among different modules, which can be further analyzed (Appendix A).

For module division, the selected TOM Type was unsigned, and mergeCutHeight was 0.9. In the cluster analysis dendrogram shown in the upper half of Appendix A, each branch represents a module, and the leaves on each branch represent a gene. Each color represents a module corresponding to the module color in the center of Appendix A. The relative expression of each gene in different tissues is shown in the lower part of Appendix A. After screening out weakly expressed genes (FPKM ≥ 5, variation of FPKM ≥ 0.5, and power = 18) in a WGCNA joint analysis, the 2549 remaining DEGs were classified into 15 modules. Different branches represent different modules, and different twigs on each branch represent a gene (Appendix A). Among the 15 modules, the genes of the cyan module (381) were the most numerous, whereas the fewest genes were classified in the gray module (51) (Appendix A).

After further analysis, we selected modules (black, blue, green-yellow, and salmon) that may be strongly associated with the salt-stress response in roots because the gene expression levels in this module increased in the 4S and 8S samples. In addition, the genes with a higher weight in each module were selected for analysis (Figure 6). Using GO and KEGG to cluster the modules, we observed that in different cluster modules, the function enrichment was strongly correlated with these modules (Figure 6). The DEGs in the black module were subjected to a GO enrichment analysis. In the biological process category, the DEGs were mainly enriched in the metabolic process, single-organism process, cellular process, response to stimulus, and other biological processes. In the cellular component category, the DEGs were mainly enriched in the membrane, membrane part, cell, and other cellular components. With regard to molecular functions, the DEGs were mainly enriched in functions such as catalytic activity, binding, and antioxidant activity. A KEGG pathway enrichment analysis showed that the DEGs were mainly enriched in metabolic processes, such as phenylpropanoid biosynthesis, biosynthesis of secondary metabolites, and metabolic pathways. In the blue, green-yellow, and salmon modules, the DEGs were subjected to a GO enrichment analysis. The DEGs were mainly enriched in the metabolic process, cellular process, and single-organism process in the biological processes category. With regard to the cellular components category, the DEGs were mainly enriched in the cell, cell part, and extracellular region. The DEGs were mainly enriched in molecular functions such as catalytic activity, binding, and nucleic acid binding transcription factor activity. A KEGG pathway enrichment analysis revealed that the DEGs were mainly enriched in metabolic processes, such as metabolic pathways, biosynthesis of secondary metabolites, and amino sugar and nucleotide sugar metabolism.

### 2.7. Gene Expression Validation by qRT-PCR Analysis

The accuracy of the transcriptome sequencing results was further verified by conducting a qRT-PCR analysis. Correlation analysis between the RNA-seq and qRT-PCR data revealed a strong correlation (R^2^ = 0.89; Figure 7), which indicated that the DEGs detected in the RNA-seq data were reliable. Based on the results of these analyses, we concluded that the overall transcriptome is altered in 8× plants compared with that of the 4× precursors.

## 3. Discussion

In this study, we observed that homologous 8× switchgrass showed enhanced tolerance to salt stress compared with that of the tetraploid precursors. The results are consistent with previous studies that reported increased tolerance to abiotic stress in other homologous polyploid plants. In a previous study, it was reported that autoploid 8× switchgrass Alamo had stronger drought tolerance than its amphidiploid 4× Alamo parent [39]. In the current study, we investigated the salt response of both 4× and 8× switchgrass at the phenotypic and physiological levels and analyzed the root transcriptome after salt treatment for 24 h. The DEGs and significantly enriched metabolic pathways in response to salt treatment were compared between the two ploidies. In addition, the WGCNA method was used to integrate the physiological and transcriptome data to further explore the possible molecular mechanisms of salt tolerance and evaluate if polyploidization is an efficient means of improving the salt tolerance of switchgrass (Figure 8).

### 3.1. Morphological and Physiological Responses of 4× and 8× Switchgrass under Salt Stress

The 4× and homologous 8× switchgrass plants showed different responses to salt treatment, of which 8× plants had superior salt tolerance. Under salt stress, the decrease in chloroplast number and damage to chloroplast membranes leads to a decrease in photosynthetic rate, leaf chlorosis, and even plant death [40]. In the present study, the leaves of 4× plants wilted earlier and yellowed more severely than those of homologous 8× plants, indicating that the chloroplasts of 4× plants were less tolerant to stress-related damage. The effect of genome doubling on salt tolerance varies from species to species [41]. In the current study, the plant height of 4× switchgrass before salt treatment was generally higher than that of homologous 8× plants but was significantly lower than that of 8× plants after salt treatment, and the leaves were severely damaged. The present results showed that 4× switchgrass was unable to adapt to salt stress and, although homologous 8× plants were also damaged to a certain extent, the symptoms were less severe than those of 4× plants. Thus, the salt tolerance of 8× switchgrass plants was enhanced. The leaves of diploid jujube wither earlier than the leaves of 4× jujube [40]. Similar results have been reported for diploid and 4× chrysanthemum [42].

Plant growth is strongly affected by saline alkali stress, which reflects the combined effects of osmotic stress, ion stress, and secondary stress [17]. The accumulation of osmoregulatory substances, such as proline and soluble sugar, is an important factor to assist in maintaining plant growth under salt stress [43]. Compared with 4× plants, the 8× switchgrass accumulated higher contents of soluble sugar and proline under salt treatment. Thus, proline and soluble sugar may play an important role in osmotic adjustment. In a saline environment, the formation of MDA is caused by the increased formation of ROS in plants and, as a result, the functioning of cell membranes is disrupted. In the present study, the MDA content of 8× plants was always lower than that of 4× plants, indicating that tetraploids were more severely affected by salt stress than 8× plants. Antioxidant enzymes are the most important components of the ROS scavenging system. SOD catalyzes the production of hydrogen peroxide and oxygen by superoxide radicals, whereas CAT is only present in peroxisomes, which can break down hydrogen peroxide into water and molecular oxygen, thereby alleviating the damage caused by salt stress [44,45]. In the current study, the activities of SOD and CAT in response to salt stress were similar and were higher in 8× switchgrass Alamo than in the 4× precursors. Salt stress causes ionic toxicity due to the accumulation of excessive detrimental ions, such as Na^+^. Therefore, plants may activate a sophisticated mechanism to exclude toxic ions to mitigate salt stress [46]. Previous studies have demonstrated that salt-tolerant genotypes may accumulate less Na^+^, which is a general rationale for combating salt stress [47,48,49]. We observed that the K^+^ content of 8× switchgrass increased significantly, and the salt tolerance was enhanced by increasing the K^+^/Na^+^ ratio, but no significant difference in the Na^+^ content of 4× is observed. This is consistent with a previous conclusion that the potassium content of diploid and tetraploid rice remains constant in response to salt stress [50]. Thus, the present results are consistent with the greater salt tolerance of 8× switchgrass Alamo.

### 3.2. Comparative Transcriptomes of 4× and 8× Switchgrass in Response to Salt Stress

In this study, we compared the transcriptomes of the roots of 8× and 4× switchgrass under the control and salt stress. The findings indicated that 8× and 4× transcriptomes changed in response to salt stress. To compare the biological functions of salt stress-responsive DETs in the two ploidies, the DET functions were annotated for each ploidy. Strikingly, a limited number of DEGs were detected in the tetraploids under the non-stress growth condition, which may be explained by the fact that the 8× plants were of autopolyploid origin. However, the number of DEGs was dramatically higher under salt stress, which underpins the importance of transcriptional reprogramming in the enhanced salt tolerance of polyploids. In addition, we noted that the number of upregulated genes in 8× Alamo was decreased in response to salt stress, which may be associated with multiple mechanisms. The GO enrichment analysis revealed that DETs specific to 4× Alamo and 8× Alamo are enriched in oxidoreductase activity and the oxidation-reduction process. In addition, enriched DEGs in 4× Alamo were involved in membrane and single-organism metabolic processes, and those in 8× Alamo were also involved in peroxidase activity and response to oxidative stress. These results suggest that the salt tolerance of polyploid switchgrass was improved by stimulation of the oxidoreductase process. This finding is consistent with a previous report that the oxidoreductase activity of autotetraploid chrysanthemum is superior to its corresponding diploid under salt treatment [51]. Under salt stress, the auto-octoploids were enriched in a greater number of salt-responsive pathways than were tetraploids. The KEGG enrichment analysis of the upregulated and downregulated DEGs showed that the octoploid response to salt stress was mainly enriched in the biosynthesis of secondary metabolites, metabolic pathways, plant MAPK signaling pathways, phenylpropanoid biosynthesis, and plant hormone signaling pathways. In addition, upregulated genes were associated with arginine and proline metabolism and starch and sucrose metabolism. The response of these DEGs with corresponding functions to abiotic stress has been reported previously in oats [29] and citrus [52].

### 3.3. Expression of Genes Associated with Plant Hormone Signal Transduction

Plant hormones play important roles in growth and developmental processes as well as biotic and abiotic stress responses [43,53,54]. In this study, the KEGG analysis of the upregulated and downregulated DEGs revealed plant hormone signal transduction pathways were enriched, involving genes that participate in the auxin, cytokinin, ABA, and ethylene pathways. It has been well documented that the signaling pathways of these four hormones play important roles in the salt stress response [55,56,57,58,59,60].

The present results indicate that the activation of hormone signaling might act as an indirect mechanism for enhanced salt tolerance by regulating other processes, such as growth and development. Auxin is involved in plant cell elongation and growth by a variety of mechanisms [61,62]. The present study revealed that genes involved in auxin biosynthesis were differentially expressed. The auxin transport inhibitor response gene *AUX1/IAA* was upregulated. *GH3* and *SAUR* are crucial genes involved in the auxin signal transduction pathway; GH3-related DEGs were upregulated, and SAUR-related genes were downregulated. Abscisic acid is the most important hormone that regulates stress responses and plays an important role in osmotic adjustment [63,64]. The ABA signal transduction pathway involves the activities of PYR/PYL, PP2C, SnRK2, and ABF. In the current study, the expression of PYR/PYL-related genes in the octoploids was downregulated under salt stress. In contrast, the expression levels of PP2C-, SnRK2-, and ABF-related genes were increased. Cytokinin is involved in a variety of physiological events, including response to environmental stress [63]. Cytokinin can regulate the adaptation of plants to salt stress during growth and development [65] and improve plant tolerance to salt stress [66]. Among the genes associated with cytokinin signaling, the expression of B-ARR genes is downregulated, indicating that cytokinin signaling may be inhibited by salt stress [67]. In the present study, the expression of A-ARR, a negative regulator of the cytokinin pathway, was increased under salt treatment, which was consistent with the aforementioned results. The difference in ploidy may cause cytokinin signal transduction to be inhibited. In addition, in 8× switchgrass, EIN3 was upregulated in the ethylene pathway. These DEGs involved in plant hormone signal transduction pathways may be directly or indirectly involved in the regulation of the plant response to salt stress.

## 4. Material and Methods

### 4.1. Plant Material, Growth Conditions, and Treatments

Switchgrass was cultivated at the Institute of Grassland, Flowers and Ecology, Beijing Academy of Agriculture and Forestry Sciences, China, under natural lighting. Octoploid materials were obtained by artificial induction of chromosome doubling in the parent tetraploid of calli by 0.04% colchicine treatment [36]. When the tetraploid and octoploid (4× and 8×, respectively) plants reached the E5 [68] developmental stage (the two-leaf stage), plants were exposed to salt stress. The control group (irrigated with half-strength (1/2) Hoagland’s nutrient solution) and the salt stress group (irrigated with 1/2 Hoagland’s nutrient solution supplemented with 150 mmol/L NaCl) were cultured in a plastic box (length × width × height: 41.0 cm × 30.5 cm × 13.5 cm) at 28 °C/20 °C (day/night) with a photoperiod of 16 h/8 h (day/night). After 7 days, two switchgrass plants (4× and 8×, respectively) with basically the same growth status were selected from each treatment, and the switchgrass plant height, stem diameter, leaf length, leaf width, and biomass were measured. All measurements were recorded with three biological replicates.

### 4.2. Measurement of Leaf Chlorophyll Content

Fresh, fully expanded upper leaves (0.5 g) were used for measurement of the chlorophyll content. Chlorophylls were extracted by soaking in 95% (*v*/*v*) acetone for 48 h in the dark. After centrifugation at 13,000× *g* for 10 min at room temperature, the absorbance of the supernatant was measured at 665 and 649 nm with a UV-2550 spectrophotometer (Shimadzu, Kyoto, Japan). The chlorophyll content was calculated using a previously published formula [69].
Chlorophyll = 5.24A_665_ + 22.24A_649_ − 15/1000 FW (mg/g FW)

### 4.3. Measurement of Leaf Soluble Sugar and Proline Contents

Leaf samples were collected after salt-stress treatment for 7 days. Samples were collected from salt-treated 4× and 8× plants and control plants at the same time. The relative water content of the second leaf for each sample was measured by weighing [70]. The soluble sugar content was determined using the Plant Soluble Sugar Content Assay Kit (BC0035, Beijing Solarbio Science and Technology, Beijing, China) in accordance with the manufacturer’s instructions. The proline content was determined using a plant proline colorimetric assay kit (Elab-E-BC-K177-S-48T, Elabscience Biotechnology Co., Ltd., Wuhan, China). Briefly, a fresh leaf sample (0.1 g) was homogenized in 1 mL of the extraction solution supplied with the kit and centrifuged at 10,000× *g* for 15 min. The supernatant was collected, and the soluble sugar and proline contents were determined in accordance with the manufacturer’s instructions.

### 4.4. Measurement of Superoxide Dismutase (SOD) and Catalase (CAT) Activities and Malondialdehyde (MDA) Content in Leaves

Leaves of switchgrass plants at the booting stage were collected and immediately frozen in liquid nitrogen. The activities of SOD and CAT and the MDA content were determined using specific assay kits (Elab-E-BC-K020, Elab-E-BC-K027-M, and Elab-E-BC-K031-M, respectively (Elabscience Biotechnology Co., Ltd.), following the manufacturer’s instructions. For each assay, a spectrophotometer (UV-1800, Shimadzu) was used to measure the absorbance.

### 4.5. Potassium (K^+^) and Sodium (Na^+^) Content in Leaves

Leaf samples were dried at 105 °C for 30 min and then at 80 °C to a constant weight. After the dried sample was ground into powder, the dry weight was recorded. The contents of K^+^ and Na^+^ were determined by inductively coupled plasma-mass spectrometry (ICP-MS; NexION-300×, PerkinElmer, Waltham, MA, USA) [39].

### 4.6. Illumina Transcriptome Library Preparation and Sequencing

After salt-stress treatment for 24 h, roots were sampled for transcriptome analysis. Root samples were collected from three individuals (representing three biological replicates) of each ploidy for each of the salt-stress treatments and control (12 samples in total) and submitted for RNA-seq by the Omicsmart Company (Guangzhou, China). For descriptive convenience, 4C and 8C represent tetraploid and octoploid samples under the control condition, and 4S and 8S represent salt-treated tetraploid and octoploid plants, respectively.

RNA library sequencing was performed on an Illumina HiSeq M2500/4000 platform by the Gene Denovo Biotechnology Co., Ltd. (Guangzhou, China). The RNA samples were detected in four steps: (1) 1% agarose gel electrophoresis was used to visualize RNA degradation and detect contaminants; (2) a NanoPhotometer spectrophotometer was used to assess RNA purity (based on the OD_260_/OD_280_ and OD_260_/OD_230_ ratios); (3) accurate quantification of the RNA concentration using a Qubit 2.0 Fluorometer; and (4) accurate detection of RNA integrity with an Agilent 2100 Bioanalyzer.

### 4.7. Transcriptome Sequencing Data Analysis

To ensure the RNA-seq data were of high quality, the raw data were filtered before analysis to reduce interference from low-quality data. First, fastp was used to control the quality of the offline raw reads and filter the low-quality data by removing reads containing adapter sequences, poly (N) sequences, poly-A tails, and low-quality reads. The subsequent analysis was based on the high-quality clean reads. The most recent version of the switchgrass genome assembly (version 4.1) was downloaded from the JGI Phytozome portal (https://phytozome-next.jgi.doe.gov, accessed on 24 April 2022). HISAT2 [71] uses global and local search methods to effectively compare split reads in RNA-seq data and is presently the most accurate software for this purpose. Thus, HISAT2 was used to perform a comparative analysis of the clean reads and the reference genome. Based on the comparison results, we used StringTie [72] to reconstruct transcripts, and RSEM [73] was used to calculate the expression amount of all genes in each sample. We first corrected the sequencing depth, then corrected the length of the gene or transcript and determined the fragments per kilobase of exon model per million mapped fragments (FPKM) value of each gene before subsequent analysis. Based on the expression results for each sample, we calculated Pearson correlation coefficients among the samples in conjunction with a principal component analysis to evaluate the repeatability among the samples and to exclude outliers. The input data for analysis of differential gene expression was the read count data obtained from the gene expression level analysis, which was analyzed with the DESeq2 [74] software. Genes were screened as significantly differentially expressed based on the criteria false discovery rate < 0.05 and |log_2_ fold change| > 1.

### 4.8. Gene Ontology (GO) Function and Kyoto Encyclopedia of Genes and Genomes (KEGG) Pathway Enrichment Analyses

For GO term enrichment analyses, we used the enrichment analysis tool accessed through the GO Consortium portal (http://geneontology.org/page/go-enrichment-analysis, accessed on 24 April 2022). The number of differentially expressed genes (DEGs) for each term was calculated to obtain the list of DEGs annotated with a GO function. A KEGG pathway enrichment analysis was performed to determine the significantly enriched biochemical, metabolic, and signal transduction pathways in which the DEGs participated.

### 4.9. Weighted Gene Co-Expression Network Analysis (WGCNA)

The WGCNA procedure is a common algorithm used for construction of a gene co-expression network, which was generated using the WGCNA R package [75]. The WGCNA algorithm is based on the assumed gene network without scale distribution, then constructs a gene co-expression correlation matrix and gene network and defines them as adjacency functions, and then analyzes and calculates the dissimilarity coefficients of different nodes to construct a hierarchical clustering dendrogram. After obtaining modules with different expression trends, gene annotation for the modules of interest was performed to explore specific molecular functions [76].

### 4.10. Quantitative RT-PCR (qRT-PCR) Analysis

Total RNA was extracted from root using the E.Z.N.A.^®^ Plant RNA Kit (Omega, China) and converted to cDNA using the PrimeScript™ RT Reagent Kit with gDNA Remover (TaKaRa, Beijing, China), following the manufacturer’s protocol. The cDNA was amplified using TB Green^®^ Premix Ex Taq™ (Tli RNaseH Plus) (TaKaRa). Quantitative RT-PCR reactions were performed using a CFX Connect Real-Time PCR Detection System (Bio-Rad, Hercules, CA, USA) with gene-specific primers listed in Table 2. The relative gene expression levels were normalized using *CYP-5* as an internal standard and calculated using the 2^−ΔΔ*C*t^ method [77] with the CFX Manager Software (Bio-Rad). Ten genes were randomly selected, and primers were designed for real-time fluorescence quantitative PCR verification (Table 2).

### 4.11. Statistical Analysis of Physiological Data

Leaves under salt-stress conditions. The assays were repeated three times with three biological replicates. The statistical significance of differences between the means was assessed using one-way analysis of variance based on a Student’s *t*-test, with *p* < 0.05 and *p* < 0.01 considered to be significant, using SPSS version XX software (SPSS Inc., Chicago, IL, USA). The OmicShare online platform (http://www.omicshare.com/tools, accessed on 21 June 2022) was used to visualize gene expression changes in salt-stress-related pathways.

## 5. Conclusions

The physiological results showed that 8× Alamo switchgrass was more strongly tolerant of salt stress than 4× plants. Two physiological reasons for the enhanced salt tolerance of 8× switchgrass are, on the one hand, reduction in MDA content and enhanced chlorophyll content, osmotic adjustment ability (indicated by soluble sugar and proline contents), and antioxidant enzyme activities (such as SOD and CAT) to better cope with oxidative stress damage; on the other hand, enhanced accumulation of K^+^ and an increase in the K^+^/Na^+^ ratio contribute to enhanced salt tolerance.

In addition, the roots of 8× plants undergo more extensive activation of hormone signaling pathways in response to salt stress, which may trigger a variety of defense machinery to counteract the salt stress. KEGG pathway analyses indicated that upregulated and downregulated DEGs were significantly enriched in plant hormone signal transduction pathways. In addition, upregulated DEGs were significantly enriched in the arginine and proline metabolism and starch and sucrose metabolism pathways. An analysis of plant hormone signal transduction pathways showed that octoploids activated the expression of genes associated with auxin (*AUX1/IAA* and *GH3*), abscisic acid (*PP2C*, *SnRK2*, and *ABF*), cytokinin (*A-ARR*), and ethylene (EIN3 receptor protein) signaling, which contribute to the superior adaptation of 8× switchgrass to salt stress. Taken together, the current study provides insights into the physiological and molecular mechanisms underlying the enhanced salt tolerance of octoploid switchgrass and will assist in the elucidation of the mechanisms of polyploids regulated in response to abiotic stresses.

## Figures and Tables

**Figure 1 plants-13-01383-f001:**
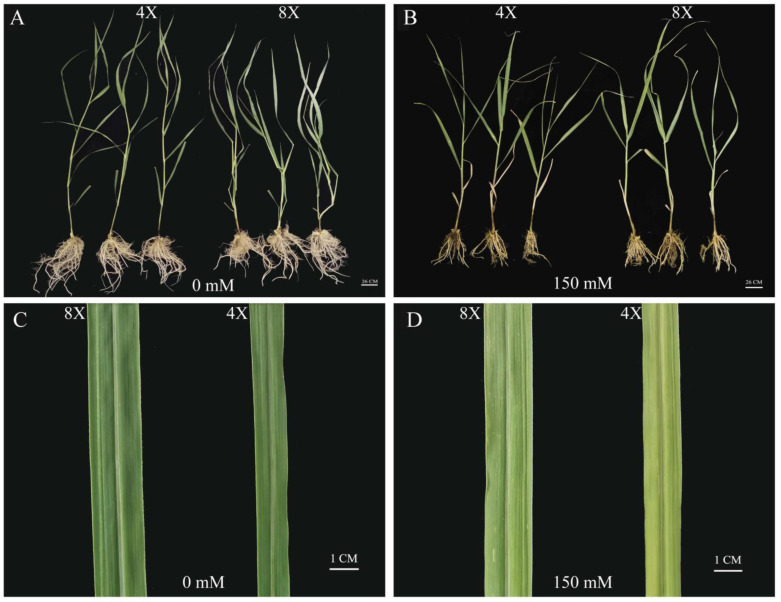
Response of tetraploid (4×) and octoploid (8×) switchgrass to salt stress. Morphological difference of 8× and 4× switchgrass under control (0 mM NaCl) (**A**) and salt (150 mM NaCl) (**B**) conditions. It shows the photographs from day 0 to the 7th day under salt stress (scale bar = 26 cm). Differences in control leaves (0 mM NaCl) (**C**) and salt (150 mM NaCl) (**D**) conditions of 4× and 8× switchgrass (scale bar = 1 cm).

**Figure 2 plants-13-01383-f002:**
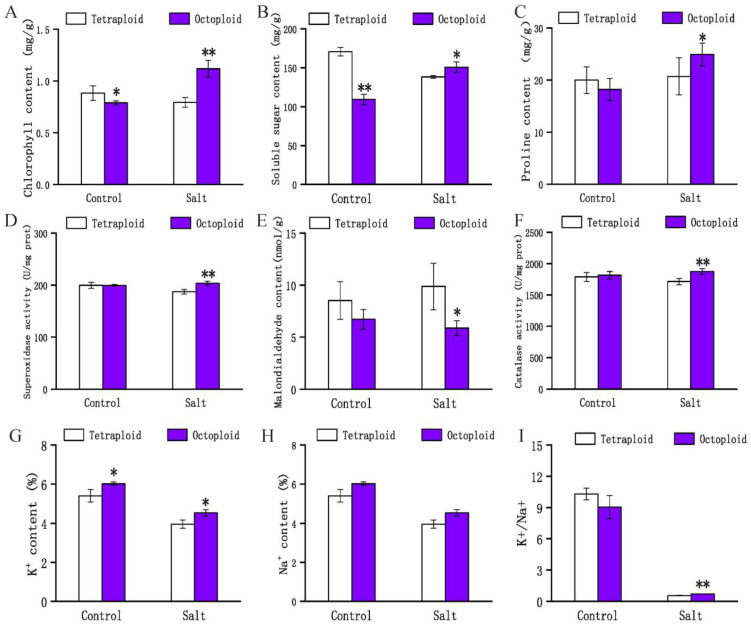
Physiological results of 4x and 8x switchgrass under control and salt stress conditions. (**A**) Total chlorophyll content. (**B**) Soluble sugar content. (**C**) Proline content. (**D**) Activity assay of superoxide dismutase (SOD). (**E**) Malondialdehyde (MDA) content. (**F**) Activity assay of catalase (CAT). (**G**) K^+^ content. (**H**) Na^+^ content. (**I**) The K^+^/Na^+^ ratio. * denotes *p* < 0.05, ** denotes *p* < 0.01. Same below.

**Figure 3 plants-13-01383-f003:**
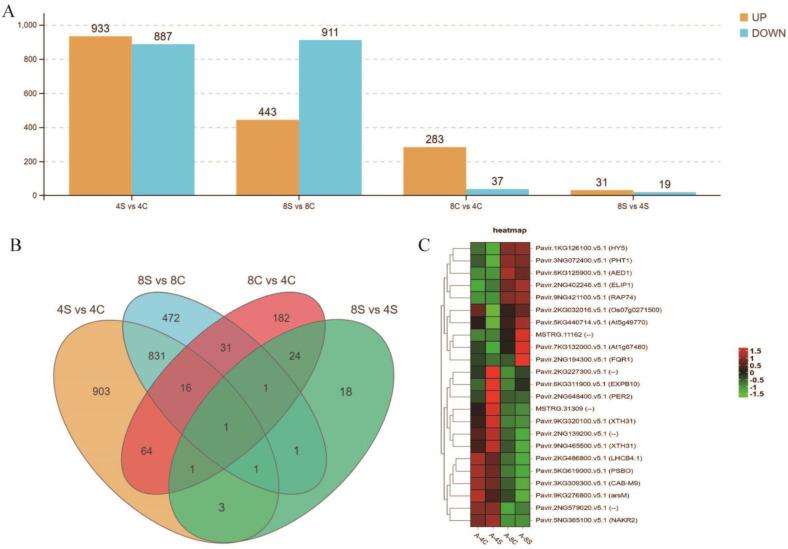
Number of significantly differentially expressed genes (DEGs) in tetraploid (4×) and octoploid (8×) switchgrass roots under control and salt conditions. (**A**) The yellow and blue bars indicate the numbers of up- and downregulated DEGs between control and salt tolerance roots in each comparison group, respectively (4S vs. 4C, 8S vs. 8C, 8C vs. 4C, 8S vs. 4S). (**B**) Venn diagram for DEGs. Each circle in the figure represents a comparison group. The numbers in the circles and overlapping regions represent the number of differential metabolites in common with the comparison group, whereas the numbers without overlaps represent the number of DEGs unique to the comparison group. (**C**) Expression heat map of DEGs 8× under salt stress (*n* = 23). Scaled values were used to present the FPKM of genes in the heat maps.

**Figure 4 plants-13-01383-f004:**
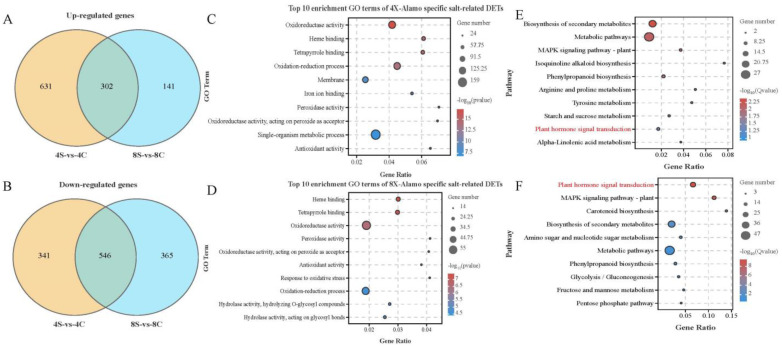
GO and KEGG enrichment analysis of tetraploid (4×) and octoploid (8×) switchgrass differentially expressed genes (DEGs) under salt stress. The salt-related up- and downregulated (**A**,**B**) DEGs Venn diagram of two ploidy. (**C**) 4× salt-related DEGs enriched top 10 GO terms. (**D**) 8× salt-related DEGs enriched top 10 GO terms. (**E**) The KEGG pathway of the top 10 upregulated genes was enriched in 302 DEGs under salt stress. (**F**) The KEGG pathway of the top 10 downregulated genes was enriched in 546 DEGs under salt stress.

**Figure 5 plants-13-01383-f005:**
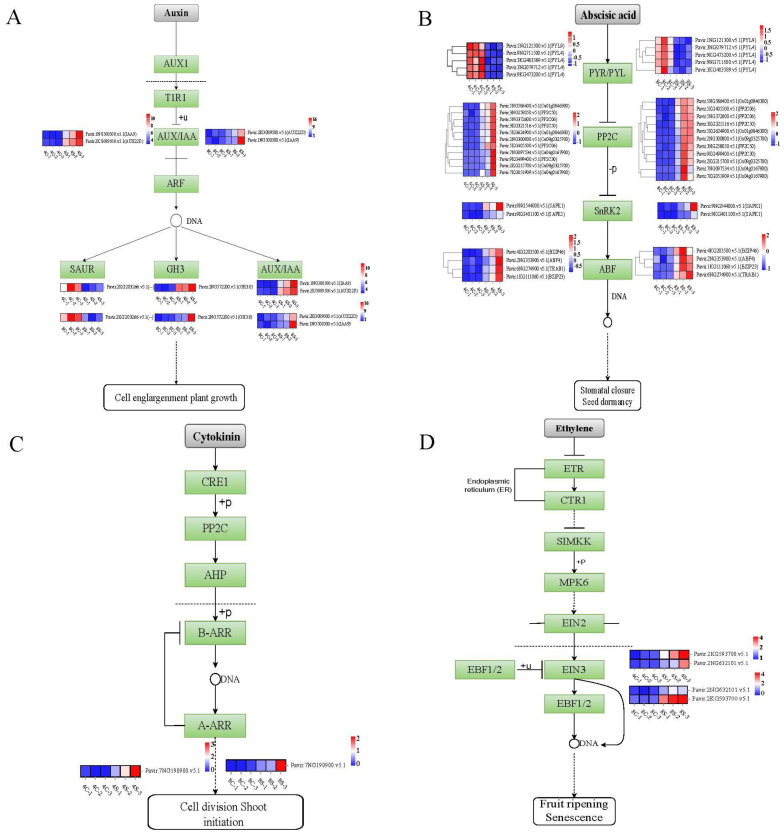
The expression pattern of key genes in hormone signal transduction pathway. (**A**) The pathway in auxin signal transduction. (**B**) The pathway in abscisic acid signal transduction. (**C**) The pathway in cytokinin acid signal transduction. (**D**) The pathway in ethylene acid signal transduction. The red rectangle indicates that the gene is enriched in the pathway. The expression data are the TPM values of the samples; red color indicates upregulated expression, and blue indicates downregulated expression.

**Figure 6 plants-13-01383-f006:**
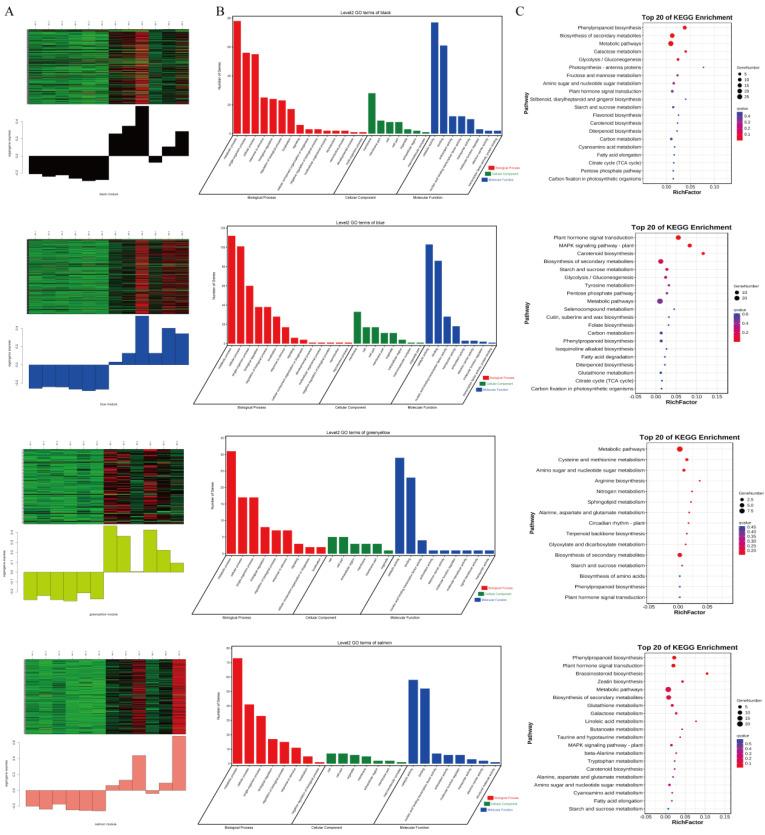
Weighted gene co-expression network analysis. (**A**) Expression heat map and expression level of genes in the black, blue, green-yellow, and salmon modules. (**B**) GO enrichment of genes in the black, blue, green-yellow, and salmon modules. (**C**) KEGG enrichment of genes in the black, blue, green-yellow, and salmon modules.

**Figure 7 plants-13-01383-f007:**
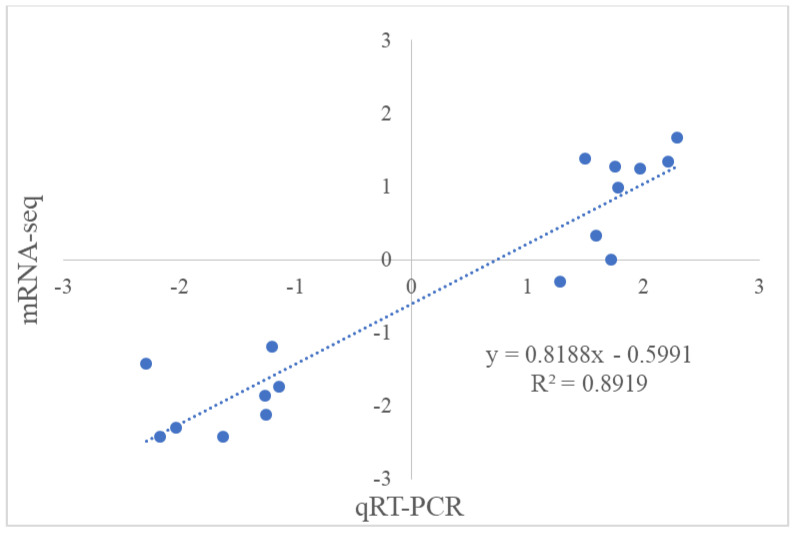
Correlation analysis of mRNA-seq and qRT-PCR.

**Figure 8 plants-13-01383-f008:**
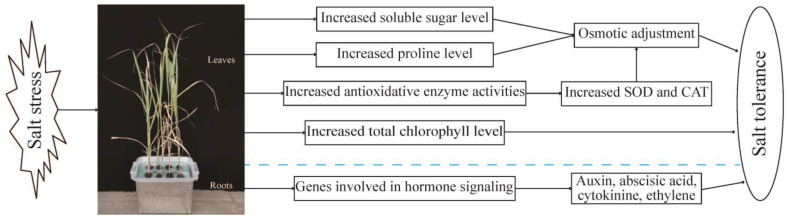
A model for mechanisms underlying the enhanced salt tolerance of octoploid (8×) switchgrass.

**Table 1 plants-13-01383-t001:** Comparison of phenotypic traits (plant height, stem diameter, leaf length, leaf width, and biomass) between octoploid (8×) and tetraploid (4×) switchgrass ‘Alamo’ under the control and salt-stress conditions. Values are the mean ± SE (*n* = 3). Values followed by different lowercase letters are significantly different (*p* < 0.05, Duncan’s multiple range test).

Ploidy	Treatments	Height (cm)	Stem Thick (mm)	Leaf Length (cm)	Leaf Width (cm)	Biomass (kg)
4×	Control	126.00 ± 2.74 a	5.62 ± 0.23 b	55.00 ± 2.12 b	1.55 ± 0.11 b	5.65 ± 0.19 a
	Salt	79.25 ± 1.92 d	4.55 ± 0.26 c	41.25 ± 1.92 c	1.30 ± 0.07 c	2.59 ± 0.17 c
8×	Control	115.25 ± 6.76 b	6.64 ± 0.30 a	69.25 ± 2.68 a	2.00 ± 0.07 a	5.45 ± 0.34 a
	Salt	86.75 ±1.48 c	5.29 ± 0.24 b	53.00 ± 3.39 b	1.60 ± 0.07 b	3.09 ±0.19 b

**Table 2 plants-13-01383-t002:** Differentially expressed genes selected for verification by quantitative RT-PCR analysis and the relevant gene-specific primers.

Gene Name	Forward Primer Sequence (5′-3′)	Reverse Primer Sequence (5′-3′)
*CYP-5* (reference gene)	CACTACAAGGGAAGCACATTCCA	TTCACCACCCCTTCCATCAC
*Pavir.4NG135600.1.v5.1*	CAATGAGCCTTGGAGTTTCA	CCAATGCTACATCCCGAATT
*Pavir.9NG711500.1.v5.1*	CGGCTCCAGAACTACCTCTC	TCCACCATGTAGGACTCCAC
*Pavir.9KG317457.1.v5.1*	ACAACATCGAGTCAGGGCAA	TCAGGTAGGGGATGATCTGC
*Pavir.7KG383000.1.v5.1*	CACATTGATGGGCAAAACTC	ATTCGACAAAGCACGAGCAG
*Pavir.9KG337100.1.v5.1*	TGGTTGACTATCCGTCTGCT	GAAAGCGTTCCATCGTAGTC
*Pavir.1NG500500.1.v5.1*	CGCTCCATAACTCCTACGAT	CAGTCCCATTCTCCATCCTC
*Pavir.2NG572200.1.v5.1*	ATTTCCTCTTCGTCAAGTCG	GGTTCTTGAAGTGGTTGCTC
*Pavir.2KG593700.1.v5.1*	GACTTCAACTTCAACACCCC	GCGTCGTACATCTCCATCAG
*Pavir.9KG401100.1.v5.1*	GGGAGGTTTATTGAGGATGA	GTTTCAGATCCCTATGGCAT
*Pavir.1KG111060.1.v5.1*	GCTATGGAGAAGGTGGTTGA	TAAGTTTTGCCACCTCAGCT

## Data Availability

Data presented in this paper are contained within the article.

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
