# Peer review of "Octoploids Show Enhanced Salt Tolerance through Chromosome Doubling in Switchgrass (*Panicum virgatum* L.)"

_plants, 2024, doi:10.3390/plants13101383_

Round 1
Reviewer 1 Report
Comments and Suggestions for Authors
The authors present a thorough overview of differences in cell response and transcriptomic differences in octoploid switchgrass versus the tetraploid progenitor. The methods are appropriate and the results clearly demonstrate and add to our knowledge of how polyploids might tolerate environmental stresses better and thus form new species or offshoots.
I have one minor comment - in the methods, the authors state that two plants with basically similar growth status were selected from each treatment for measurement, but then say that three biological replicates were performed... surely they needed three plants per treatment for this to be the case?
Author Response
Thank you very much for taking the time to review this manuscript. And thank you for pointing this out. Two switchgrass plants mean octoploi and tetraploid switchgrass. All measurements were recorded with three biological replicates.
Reviewer 2 Report
Comments and Suggestions for Authors
The reviewed publication is well-written, clear, and understandable. Its individual stages together form a cohesive whole. My main concern pertains to editorial matters - please revise the manuscript according to the guidelines for the Plants journal (the Materials and Methods section should come after the Discussion). Figures added as supplements should not appear in the main text. Please try to increase the resolution of figures with a large amount of data and text.
Author Response
Thank you very much for taking the time to review this manuscript. And thank you for pointing this out. We agree with this comment. According to the guidelines for the Plants journal, we take the Materials and Methods section come after the Discussion.
Reviewer 3 Report
Comments and Suggestions for Authors
Dear Author
The article titled "Octoploids Show Enhanced Salt Tolerance through chromosome doubling in Switchgrass (Panicum virgatum L.)" was evaluated in terms of scientific content and writing format. The scientific content of the article was found to be high. Although the writing style of the article is generally appropriate, some minor additions and corrections need to be done. The necessary additions and corrections are shown on the annotated manuscript.

Dear Author
Dear Editor
The article titled "Octoploids Show Enhanced Salt Tolerance through Chromosome Doubling in Switchgrass (Panicum virgatum L.)" was valuated in terms of scientific content and writing format. The scientific content of the article was found to be high. Although the writing style of the article is generally appropriate, some minor additions and corrections need to be done. The necessary additions and corrections are shown in the annotated manuscript.
Author Response
Thank you very much for taking the time to review this manuscript. And thank you for pointing this out. We agree with this suggestion and have modified the terminology throughout the text as appropriate. Thanks for your kind reminder. We have corrected it in the manuscript. The manuscript has been carefully checked and typo errors have been corrected. Please check the annexed file to see the new manuscript.